# The Insertion Domain of Mti2 Facilitates the Association of Mitochondrial Initiation Factors with Mitoribosomes in *Schizosaccharomyces pombe*

**DOI:** 10.3390/biom15050695

**Published:** 2025-05-10

**Authors:** Ying Luo, Jürg Bähler, Ying Huang

**Affiliations:** 1Jiangsu Key Laboratory for Microbes and Genomics, School of Life Sciences, Nanjing Normal University, 1 Wenyuan Road, Nanjing 210023, China; ying.luo@ucl.ac.uk; 2Institute of Healthy Ageing, Department of Genetics, Evolution & Environment, University College London, London WC1E 6BT, UK

**Keywords:** fission yeast, mitochondrial translation, translation initiation factor, insertion domain

## Abstract

Translation initiation in mitochondria involves unique mechanisms distinct from those in the cytosol or in bacteria. The *Schizosaccharomyces pombe* mitochondrial translation initiation factor 2 (Mti2) is the ortholog of human MTIF2, which plays a vital role in synthesizing proteins in mitochondria. Here, we investigate the insertion domain of Mti2, which stabilizes its interaction with the ribosome and is crucial for efficient translation initiation. Our results show that the insertion domain is critical for the proper folding and function of Mti2. The absence of the insertion domain disrupts cell growth and affects the expression of genes encoded by mitochondrial DNA. Additionally, we show that Mti2 physically interacts with the small subunits of mitoribosomes (mtSSU), and deletion of the insertion domain dissociates mitochondrial initiation factors from the mitoribosome, reducing the efficiency of mitochondrial translation. Altogether, these findings highlight the conserved role of the insertion domain in facilitating translation initiation in fission yeast and thus reveal shared principles of mitochondrial translation initiation in both fission yeast and humans.

## 1. Introduction

The mitochondria are fundamental organelles in most eukaryotic cells, playing an essential role in producing cellular energy through the oxidative phosphorylation system (OXPHOS). In addition, they are also involved in a wide range of metabolic pathways for amino acids, nucleotides and lipids, as well as the regulation of apoptosis and aging [1,2]. Given their central role in energy homeostasis and metabolic regulation, the dysfunction of mitochondria is associated with a variety of human diseases, ranging from neurodegenerative disorders to metabolic syndromes [3,4].

Mitochondria contain their own DNA (mtDNA), which encodes essential components required for mitochondrial gene expression and OXPHOS. In the fission yeast *Schizosaccharomyces pombe* (*S. pombe*), the ~19 kb mtDNA encodes seven core subunits of OXPHOS proteins, including Cob1 (subunit of complex III, also called Cytb), Cox1, Cox2 and Cox3 (subunits of complex IV), Atp6, Atp8 and Atp9 (subunits of ATP synthase). Additionally, the mtDNA encodes the mitochondrial ribosomal protein Var1, the RNA subunit of RNaseP (*rnpB*), two rRNAs (*rns* and *rnl*) and 25 tRNAs. Unlike human mtDNA, both *S. pombe* and *Saccharomyces cerevisiae* (*S. cerevisiae*) mtDNAs lack complex I but instead encode alternative NADH dehydrogenases [5]. This distinct genomic organization highlights the evolutionary diversity of mitochondrial genomes and their functional adaptation to diverse cellular environments.

The mechanism of mitochondrial translation shares greater similarity with bacterial translation than with its cytosolic counterpart, as mitochondria are derived from bacteria [6,7]. Despite this evolutionary relationship, mitochondrial translation initiation also exhibits features that are distinct from bacterial systems. In bacteria, the translation initiation requires three essential initiation factors (IF1, IF2 and IF3), which facilitate the binding of the initiator to the mitochondrial ribosome. During the initiation of translation, the small ribosomal subunit, together with the initiator tRNA, assembles on the mRNA transcript. The small subunit contains three functionally distinct tRNA-binding sites: the amino acid (A) site, which accommodates incoming aminoacyl-tRNAs; the polypeptide (P) site, which holds the tRNA carrying the growing polypeptide chain; and the exit (E) site, through which deacylated tRNAs exit the ribosome [8]. In contrast, translation initiation in mitochondria involves only two initiation factors (mtIF2 and mtIF3), as IF1, a universally essential component in bacterial and cytosolic translation, is notably absent in mitochondria [9,10,11,12]. Another difference between bacterial and mitochondrial translation initiation is the formylation of initiator tRNA. In *E. coli*, the initiation of translation depends on a formylated initiator tRNA (fMet-tRNA_i_^fMet^), whereas, in mitochondria, translation initiates with non-formylated initiator tRNA. Like in mammals, the formylation of Met-tRNA_i_^fMet^ in *S. cerevisiae* enhances its affinity with mtIF2 [13]. Another distinction in mitochondrial translation is the involvement of translational activators, which function in conjunction with initiation factors (IFs) to regulate mitochondrial gene expression. These activators play a crucial role in facilitating mitochondrial translation and often act in a mRNA-specific manner [14]. In *S. cerevisiae*, for instance, Sov1 is required for the translation of *VAR1*. While Cbs1, Cbs2, Cbp1 and Cbp3-Cbp6 form a complex involved in the translation of *COB*, Pet309 and Mss51 are critical for the translation of *COX1*, ensuring efficient synthesis of its encoded protein [6].

Despite possessing a conserved core fold, mitochondrial translation initiation factor 2 (mtIF2) is different from bacterial IF2 in several functional domains. A 37-amino acid insertion domain, located between domains II and III, was identified through a cryo-EM structure study of the entire translation initiation complex from mammalian mitochondria; this domain extends toward the decoding center and adopts the form of an α-helix, stabilizing the binding of the leaderless mRNAs and causing conformational changes in the rRNA nucleotides [15]. It has been suggested that the insertion domain of mtIF2 in vertebrate mitochondria performs a similar role to that of bacterial IF1 [16,17]. This insertion domain was first identified in bovine mtIF2, where mutations in this region dramatically impair the formation of the initiation complex and disrupt its association with the small subunit of the mitoribosome [18,19]. Similarly, in human MTIF2, this domain has been shown to enhance the translation initiation efficiency by stabilizing the mitoribosome association and ensuring accurate start codon selection [15]. The evolutionary and genetic analysis of mitochondrial translation initiation factors suggests that the mtIF2 insertion domain functionally compensates for the universal absence of IF1 in mitochondria and exhibits high variability in length without perturbing protein function and primary sequence conservation across vertebrates. Sequence analysis of the mitochondrial translation initiation factors further demonstrates that the insertion domain has a strong bias toward the amino acid composition, particularly for glutamate and lysine. In the human MTIF2 insertion domain, glutamate and lysine account for 20.3% and 21.9%, respectively, a pattern also observed in fungal homologs, such as in *S. pombe* and *S. cerevisiae* [20].

To investigate whether the insertion domain plays a similar functional role in fungal mitochondria, we investigated the insertion domain of Mti2 (the mtIF2 homolog in *S. pombe*). We predicted the structure of Mti2 both with and without the insertion domain and performed structure alignment, suggesting that the insertion domain is essential for the proper folding of Mti2. Additionally, functional assays demonstrated that the insertion domain is critical for mitochondrial function and the translation of mtDNA-encoded genes. Furthermore, we explored the physical interaction between Mti2 and the mitochondrial ribosomal subunits. The coimmunoprecipitation results confirmed that Mti2 physically interacts with the small subunits of mitoribosomes (mtSSU) rather than large subunits of mitoribosomes (mtLSU). Sucrose sedimentation analysis further indicated that the absence of the insertion domain disrupts the association of mitochondrial translation initiation factors to mtSSU, as well as the assembly of the mitoribosome. These findings suggest that, similar to its role in mammalian mitochondria, the insertion domain of Mti2 in *S. pombe* plays a conserved role in promoting translation initiation by facilitating mitoribosome association and thus reveals the shared principles of mitochondrial translation initiation in both fission yeast and humans.

## 2. Materials and Methods

### 2.1. Strains and Media

The *S. pombe* strains used in this study are listed in Appendix A. The Δ*mti2* strain was generated by homologous recombination using pFA6a-kanMX6 [21]. The Δ*mti2* strain was verified by PCR using check primers. The schematic view of the construction of the *mti2* deleting insertion domain (aa443-477) strain (referred to as the *mti2*Δ*insertion* strain) is shown in Appendix A. Briefly, the fragments of Mti2^1–^^442^ and Mti2^478–686^ were amplified by PCR; these two fragments were fused into one single fragment using overlapping PCR and subsequently integrated into XbaI/SmaI sites of pJK148 [22]. The plasmid was then linearized with NruI and transformed into wild-type strain yHL6381. A strain expressing C-terminal-tagged Mti2 was generated by integrating the corresponding tagging cassette into the endogenous *mti2* locus in strain yHL6381. The tagging cassette was generated via overlapping PCR. The *mti2*Δ*insertion* and Mti2-FLAG strains were verified by Western blotting with the primary antibody against Mti2 and FLAG, respectively.

*S. pombe* cells were cultured in rich Yeast Extract with Supplements medium (YES: 3% glucose, 0.5% yeast extract, 225 mg/L adenine, histidine, leucine, uracil and lysine hydrochloride). The described standard protocols of *S. pombe* were adhered to in this study [23].

### 2.2. Prediction and Alignment of Protein Structure

The structure prediction of *S. pombe* Mti2 and Mti2Δinsertion was conducted by AlphaFold 3 [24] using the default parameters (https://alphafoldserver.com/, accessed on 25 November 2024). To evaluate whether the deletion of the insertion domain altered the structure of Mti2, TM-align was used for the comparison of protein structures and alignment [25]. TM-align provides a TM-score to quantify structural similarities, with a TM-score <0.3 indicating random structural similarity, while scores ranging from 0.5 to 1.0 suggest that the two structures share a very similar fold. In contrast, a higher root mean square deviation (RMSD) value reflects a significant structural difference between the compared models.

### 2.3. Real-Time Quantitative Yeast Growth Assays

The quantitative growth assays were performed using FlowerPlates in a BioLector Microbioreactor (m2p-labs, Baesweiler, Germany), as described in [26]. Briefly, the wild type, *mti2*Δ*insertion*, and Δ*mti2* strains were precultured overnight in YES medium at 32 °C with 180 rpm shaking, and the cultures were then diluted to an initial OD_600_ of 0.2 with fresh YES medium and grown for ~4 h to the mid-exponential phase. Subsequently, the cultures were further diluted to achieve an initial OD_600_ of 0.02, and 1.5 mL cultures were incubated in triplicate at 32 °C with 85% humidity and 1000 rpm shaking. The growth was measured every 10 min until the cells reached the stationary phase. The growth data were normalized to the initial time point (time 0) with the R package *shiny* (version 1.10.0, Posit, PBC, Boston, MA, USA, https://CRAN.R-project.org/package=shiny, accessed on 31 December 2024), and the mean growth curves were analyzed using *grofit* [27]. Statistical analysis was performed using one-way ANOVA by R (version 4.4.1, R Foundation for Statistical Computing, Vienna, Austria) [28,29].

### 2.4. Quantitative Real-Time RT-PCR

The wild type, *mti2*Δ*insertion* and Δ*mti2* strains were precultured in YES overnight at 32 °C and then diluted to an initial OD_600_ of 0.2 in fresh YES. Cells were collected after culturing for 6 h, and total RNA was extracted using an E.Z.N.A. Yeast RNA Kit (OMEGA BIO-TEK, Norcross, GA, USA), followed by reverse transcription using the HiScript III RT SuperMix for qPCR (Vazyme, Nanjing, Jiangsu, China). qPCR was then performed using the Taq Pro Universal SYBR qPCR Master Mix (Vazyme, Nanjing, Jiangsu, China). All reactions were performed in triplicate. The Ct values were normalized against the *act1* (actin) levels. Fold changes in the mRNA levels of the *mti2*Δ*insertion* and Δ*mti2* strains in comparison to the wild type were analyzed by the 2^−^^∆∆Ct^ method. GraphPad Prism (version 10.1.1, GraphPad Software, Boston, MA, USA) was used to determine the *p*-values. The primers for qPCR are listed in Appendix A.

### 2.5. Isolation of Mitochondria and Western Blot Analysis

Crude mitochondria from *S. pombe* were isolated as previously described [30]. Briefly, cells were precultured overnight at 32 °C in YES medium, harvested and washed with sterile water. The cell walls were enzymatically digested using lysis enzymes derived from *Trichoderma harzianum* (Sigma-Aldrich, St. Louis, MO, USA) for 30 min to generate protoplasts. The protoplasts were then disrupted using a Dounce tissue grinder (Sigma-Aldrich, St. Louis, MO, USA). The lysate was subjected to centrifugation at 5000 rpm for 5 min at 4 °C, and the supernatant was collected. The low-speed centrifugation step was repeated 3 to 4 times, until no visible cell debris remained. The resulting supernatant was then centrifuged at maximum speed (12,000 rpm) for 20 min to pellet the crude mitochondria. The resulting mitochondrial pellet was resuspended in SH buffer (0.6 M sorbitol and 20 mM HEPES, pH 7.5) and stored at −80 °C for further use. The mitochondrial proteins were detected by Western blotting with the corresponding primary antibodies. The primary antibodies against anti-Cob1 (aa 256-268), anti-Cox1 (aa 524-537), anti-Cox2 (aa 149-162), anti-Cox3 (aa 123-136), anti-Atp6 (aa 2-21), anti-Mti2 (aa 667-686), anti-Mti3 (aa 214-233), anti-Mrp5 (aa 368-387), anti-Rsm24 (aa 239-258) anti-Mrpl16 (aa 37-54) and Mrpl40 (aa 32-50) were prepared as described [31]. Hsp60 served as the loading control. Original figures can be found in Appendix A.

### 2.6. Immunoprecipitation Assay

The wild-type and Mti2-FLAG strains were precultured overnight at 32 °C in YES medium, diluted to an initial OD_600_ of 0.2 in fresh YES and grown to the mid-exponential phase. Cells were harvested after 10 h of culturing and suspended in binding buffer (20 mM Tris-HCI, pH 8.0, 150 mM NaCl, 10% glycerol, 1% Nonidet P-40 and 1 mM PMSF). The cells were then homogenized with glass beads (Sigma-Aldrich, St. Louis, MO, USA) using the FastPrep-24 instrument (MP Biomedicals, Irvine, CA, USA) at a speed of 6 M/S for 20 s per cycle, repeated for 10 cycles with 5 min of incubation on ice between each cycle. The lysates were then centrifuged at 12,000 rpm for 20 min at 4 °C to remove cell debris. Then, 100 μL of supernatant was collected as the input (IN) sample and mixed with 25 μL of the protein loading buffer (5×) and boiled at 100 °C for 5 min. The remaining supernatant was incubated with anti-FLAG beads (Sigma-Aldrich, St. Louis, MO, USA) overnight at 4 °C with a gentle rotation. After incubation, the beads were pelleted by centrifugation at 500× *g* for 3 min, and 100 μL of supernatant was collected as the supernatant (S) sample, while the remaining supernatant was discarded and the beads were washed three times with washing buffer (10 mM Tris-HCI, pH 8.0, 68.5 mM NaCl, 5% glycerol and 0.1% Nonidet P-40) as described [32]. Protein loading buffer (1×) was used to elute the bound proteins (IP) and boiled at 100 °C for 5 min. All collected samples (IN, S and IP) were analyzed by Western blotting using specific antibodies. The experiment was independently repeated three times.

### 2.7. Sucrose Gradient Sedimentation Analysis

Sucrose gradient sedimentation analysis was conducted as previously described [33,34]. Briefly, 2 mg of mitochondria were suspended in 300 μL of lysis buffer containing 20 mM HEPES (pH 7.5), 100 mM KCl, 10 mM MgCl_2_, 1% digitonin, 1 mM PMSF and EDTA-free protease inhibitor (Roche, Basel, Switzerland). The suspension was incubated on ice for 25 min with gentle inversion every 5 min to improve the lysis efficiency. The lysate was collected by centrifugation at 12,000 rpm for 20 min at 4 °C. The resulting supernatant was then loaded onto the top of a 10–34% sucrose gradient solution (20 mM HEPES, pH 7.5, 100 mM KCl, 10 mM MgCl_2_, 0.1% digitonin, 1 mM PMSF and EDTA-free protease inhibitor (Roche, Basel, Switzerland)) in ultracentrifugation tubes (#328874, Beckman Coulter, Brea, CA, USA) and subjected to ultracentrifugation using a SW 60 Ti rotor (Beckman Coulter, Brea, CA, USA) at 40,000 rpm for 3 h at 4 °C. Twelve fractions (~350 μL per fraction) were collected from the bottom to the top. The proteins were precipitated by adding an equal volume of 50% trichloroacetic acid (TCA), followed by overnight precipitation at −80 °C. The resulting protein pellets were collected by centrifugation at 12,000 rpm for 20 min at 4 °C and were then washed with 1 mL of ice-cold acetone, suspended in 1× protein loading buffer and detected by Western blotting using the corresponding primary antibodies. The experiment was independently repeated three times.

## 3. Results

### 3.1. The Insertion Domain Is Required for the Proper Folding of Mti2

To determine the insertion domain in *S*. *pombe* Mti2 that corresponds to the insertion domain of human MTIF2, a sequence alignment was performed between human MTIF2 and *S. pombe* Mti2 (Figure 1a). The domains of human MTIF2 were annotated following the framework described previously [15], and the corresponding domains in *S. pombe* Mti2 were identified (Figure 1b). The G-domain is a structurally and functionally conserved domain responsible for GTP binding and hydrolysis and plays a critical role in regulating protein synthesis during translation initiation, elongation and termination [35]. Domain II promotes initiator tRNA association with the small ribosomal subunit in a GTP-dependent manner during translation initiation [36]. Domain III adopts a conserved *α*/*β*/*α* structural fold, featuring a core parallel β-sheet that contributes to the proper binding of the initiator tRNA to the P site [37,38]. Domain IV, characterized by a *β*-barrel fold, facilitates conformational changes between the GTP- and GDP-bound states [38]. Like in human MTIF2, the insertion domain in Mti2 is located within the linker region between domain II and domain III (Figure 1b). However, the sequence alignment indicates that the insertion domains of human MTIF2 and *S. pombe* Mti2 exhibits low sequence similarity. This finding is consistent with the observation that the insertion domain varies greatly between species and is restricted in length [20].

To explore the potential functional significance of the insertion domain in Mti2, we predicted the tertiary structures of Mti2 both with and without the insertion domain using AlphaFold 3 [24]. The ribbon patterns of the predicted tertiary structures revealed that the absence of the insertion domain resulted in significant alterations in protein folding. Using domain IV as a reference, notable shifts in the spatial arrangement of other domains were observed (Figure 1c,d). Both AlphaFold 3 predictions showed high predicted Local Distance Difference Test (pLDDT) scores, indicating high model confidence. The AlphaFold 3 predictions in Figure 1c,d are colored by domains, while the same models are colored according to pLDDT scores in Appendix A. The structural alignment of Mti2 with and without the insertion domain was subsequently performed by TM-align (Figure 1e). The analysis yielded a TM-score of 0.47 and an RMSD of 3.93 Å, indicating a low degree of structural similarity. These structural predictions suggest that the absence of the insertion domain significantly disrupts the proper folding of Mti2, supporting its important role in Mti2 function.

### 3.2. Deletion of the Insertion Domain Impairs Mti2 Function

To further investigate whether the deletion of the insertion domain affects the normal function of Mti2, we constructed a deletion strain of the entire *mti2* gene (Δ*mti2*) and a deletion strain of the insertion domain only (*mti2*Δ*insertion*; Appendix A). We examined whether these mutant strains impacted the growth of proliferating cells using a Biolector microbioreactor. The results revealed that both the *mti2*Δ*insertion* and Δ*mti2* cells exhibited reduced growth rates and prolonged lag phases compared to the control cells (Figure 2a). A spotting assay was then performed to assess cell growth under different carbon-source conditions. In glucose media, *S. pombe* cells grow by fermentation with low mitochondrial respiratory demand, while, in glycerol media, cells require high mitochondrial respiratory activity [41]. Compared to wild-type cells, the growth of *mti2*Δ*insertion* and Δ*mti2* cells was only marginally reduced in glucose media but was significantly impaired in glycerol media (Figure 2b). This result suggests that mitochondrial respiration is similarly inhibited in both mutants.

Mti2 is required for mitochondrial protein synthesis and normal levels of mtDNA-encoded proteins [31]. To assess whether the insertion domain is required for the expression of mtDNA-encoded genes, we examined both the RNA and protein levels of these genes in *mti2*Δ*insertion* cells using qPCR and Western blotting, respectively. We found that *mti2*Δ*insertion* cells, similar to Δ*mti2* cells*,* showed significantly reduced RNA levels of core subunits of respiratory chain complexes, including *cob1* (complex III), *cox1*, *cox2* and *cox3* (complex IV) and *atp6*, *atp8* and *atp9* (ATP synthase). However, the RNA level of *var1*, which encodes a mitochondrial ribosomal protein, remained stable in both mutants (Figure 2c). Furthermore, both *mti2*Δ*insertion* and Δ*mti2* cells showed nearly abolished expression of the corresponding OXPHOS-related proteins (Figure 2d). Collectively, these findings highlight the critical role of the insertion domain for Mti2 function and mitochondrial gene expression.

### 3.3. Mti2 Physically Interacts with the Small Mitoribosome Subunit

Previous studies have shown that Mti2 co-sediments with mtSSU in the same fractions using sucrose sedimentation analysis, suggesting its potential association with mtSSU [31]. To investigate whether Mti2 physically interacts with the mtSSU, we generated a Mti2-FLAG tagged strain and performed co-immunoprecipitation (Co-IP) assays. Our results revealed that the mtSSU, but not the mtLSU, specifically co-precipitated with Mti2-FLAG (Figure 3). Moreover, no direct physical association was detected between the two mitochondrial translation initiation factors Mti2 and Mti3. This finding provides direct evidence of a physical interaction between Mti2 and the mtSSU, reinforcing the model that Mti2 plays a direct role in mitochondrial translation initiation by associating with the mitoribosomes.

### 3.4. Deletion of the Insertion Domain Reduces the Affinity Between Mti2 and Mti3 and the mtSSU

Previous studies have shown that the insertion domain of human MTIF2 exhibits functional similarities to bacterial IF1, despite its distinct structural fold. The deletion of the insertion domain of MTIF2 leads to impaired translation efficiency in vitro [15]. It is thought that the insertion domain facilitates mitochondrial translation by preventing premature entry of elongation factor tRNAs into the A-site and minimizing mRNA slippage, thereby ensuring accurate reading frame selection [15].

To explore whether the insertion domain in *S. pombe* plays a similar role in the translation initiation, we conducted sucrose sedimentation analysis using wild-type, *mti2*Δ*insertion* and Δ*mti2* cells. This approach enabled us to assess the effect of the insertion domain on the interaction between the mitochondrial translation initiation factors and mitoribosomes. In wild-type cells, the mtSSU, mtLSU and mitoribosome complexes were enriched in distinct fractions, with the mitochondrial initiation factors co-sedimented with the mtSSU (Figure 4a). Notably, deletion of the insertion domain resulted in a greatly decreased association of the translation initiation factors Mti2 and Mti3 with the mtSSU. In the *mti2*Δ*insertion* strain, the mtSSU marker Mrp5 peaked in fractions 6–7, while the initiation factors Mti2 and Mti3 mainly peaked in fractions 4–5 (Figure 4b). However, there was substantial overlap between the initiation factors and the mtSSU fractions, suggesting that the insertion domain deletion disrupts, but not completely abolishes, the association with mtSSU. Moreover, the deletion significantly reduced the association with assembled mitoribosomes and impaired mitoribosome assembly. Specifically, a distinct mitoribosome peak was observed in fraction 11 in the wild-type cells (Figure 4a) but was markedly diminished in the *mti2*Δ*insertion* cells (Figure 4b), indicating a defect in mitoribosome assembly. This phenotype is similar to that observed in Δ*mti2* cells (Figure 4c). These findings indicate that, similar to MTIF2 in humans, the insertion domain in *S. pombe* Mti2 promotes the efficiency of translation initiation by facilitating the association of initiation factors with the mitochondrial ribosome. We conclude that the insertion domain plays a conserved and crucial role in the function of MTIF2/Mti2 in mitochondrial translation in both fission yeast and humans, species that have diverged over a billion years ago.

## 4. Discussion

Mitochondrial translation initiation is a complex and highly regulated process that diverges significantly from its bacterial and cytosolic counterparts. While sharing a core mechanism with bacterial translation, mitochondria have evolved unique adaptations, including the use of methionyl-tRNA for both initiation and elongation and the absence of Shine–Dalgarno sequences, a conserved ribosome-binding site in prokaryotic mRNAs that aligns the ribosome with the start codon to initiate translation [42]. Another distinctive feature is the absence of IF1, which is universally present in other translation systems and increases the accuracy of the usage of tRNAs during elongation [43]. The insertion domain of mtIF2 has been proposed to compensate for the absence of IF1 in vertebrates, suggesting that they may share a similar function in diverse eukaryotes [15]. Phylogenetic and sequence analyses of mitochondrial translation initiation factors 2 showed that the insertion domain is only conserved in mammals [18,44], and full-length insertion domains are limited to vertebrates [20]. However, whether the insertion domains play a similarly conserved role in other eukaryotes, particularly fungi, remains poorly understood. In this study, we investigated the functional significance of the insertion domain in Mti2, the mtIF2 homolog in *S. pombe.* Our findings demonstrate that the insertion domain is crucial for Mti2 protein folding, mitochondrial respiration and association of the initiation factors with mitoribosomes. Additionally, we show that the deletion of the insertion domain affects the expression of mtDNA-encoded genes. Both *mti2*Δ*insertion* and Δ*mti2* cells exhibited a general reduction in the RNA levels of OXPHOS-related genes and rRNAs (*rns* and *rnl*) (Figure 2c), rather than specifically affecting a complex or a protein. Notably, the RNA level of *var1*, encoding a mitochondrial ribosomal subunit, remained relatively stable. This pattern suggests that not all mtDNA-encoded genes are equally affected. The decrease in mRNA levels was generally consistent with the decrease in the corresponding protein levels. The protein levels of these mtDNA-encoded genes were almost completely abolished in the *mti2*Δ*insertion* and Δ*mti2* strains (Figure 2d), indicating that the deletion of the insertion domain not only reduces transcript abundance but also impairs mitochondrial translation. This conclusion is further supported by the sucrose sedimentation analysis (Figure 4). Together, these findings support the hypothesis that the insertion domain plays an important and conserved role in mitochondrial translation initiation.

We show that Mti2 physically interacts with the mtSSU to facilitate the initiation of mitochondrial protein translation (Figure 3). Deletion of the insertion domain disrupts the association between mitochondrial initiation factors (Mti2 and Mti3) and the mtSSU (Figure 4b). In human MTIF2, the deletion of the Trp-Lys-X-Arg motif, which is part of the insertion domain, severely impairs its function, suggesting that the insertion domain enhances the efficiency of translation initiation. This disruption is likely due to the loss of the insertion domain’s role in preventing the premature binding of elongator tRNAs to the A-site and minimizing mRNA slippage to ensure accurate reading frame selection [15]. According to the sequence alignment, the insertion domains share similar physicochemical properties that likely underlie their functional roles in mitochondrial translation initiation. However, the Trp-Lys-Lys-Arg motif in human MTIF2 and the Gln-Glu-Glu-Thr motif in *S. pombe* Mti2 feature distinct amino acid sequences (Figure 1a). Both motifs consist of highly polar and charged residues, which likely facilitate the interactions with ribosomal RNA and/or other components of the translational machinery, thereby ensuring the efficient translation of leaderless mRNAs. Similarly, despite the low sequence identity of the insertion domains across species (Figure 1a), their deletion results in impaired mitochondrial translation in different eukaryotes, highlighting their conserved functional importance.

This study provides fresh insights into the structural and functional role of the insertion domain in mitochondrial translation initiation, demonstrating its essential contributions to Mti2 stability, mitoribosome association and translation efficiency in *S. pombe*. By establishing functional parallels between fungal and vertebrate mtIF2 insertion domains, our findings highlight a conserved adaptive mechanism, compensating for the absence of IF1 in mitochondrial translation. These results broaden our understanding of the evolutionary divergence and conservation of mitochondrial translation mechanisms across eukaryotic species.

## 5. Conclusions

Our study demonstrates that the insertion domain of Mti2 plays a conserved and crucial role in Mti2 function and mitochondrial translation. This domain is required for the proper protein folding of Mti2, mitochondrial respiration and efficient mitochondrial translation. Moreover, we show that Mti2 physically interacts with the mtSSU. Although the insertion domain of mtIF2 has been thought to be primarily conserved in vertebrates, our findings reveal that it performs a similar function in fission yeast, compensating for the absence of IF1 and facilitating translation initiation, which suggests a more widespread functional conservation across eukaryotes.

## Figures and Tables

**Figure 1 biomolecules-15-00695-f001:**
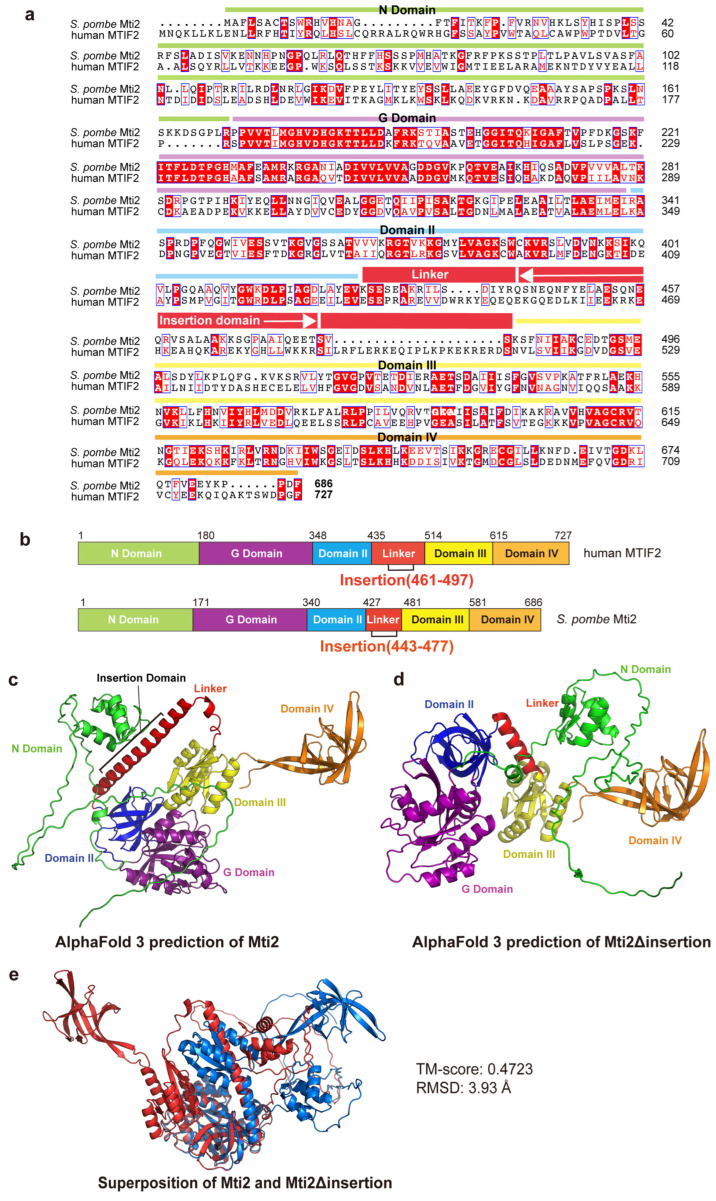
The insertion domain plays a crucial role in the proper protein folding of Mti2. (**a**) Sequence alignment of *S. pombe* Mti2 and human MTIF2; the sequences were obtained from GenBank, and sequence alignment was conducted by Clustal Omega [39]. The domains of human MTIF2 were annotated following the framework described [15], and the corresponding domains in *S. pombe* Mti2 were identified through sequence alignment and are highlighted with different colors. The location of the insertion domain of Mti2 is indicated by white arrows. Sequence similarities were rendered using ESPript 3.0 [40]. (**b**) Schematic representation of domain organization of human MIIF2 (top) and *S. pombe* Mti2 (bottom) according to sequence alignment. (**c**,**d**) Ribbon representations of the predicted three-dimensional structure of *S. pombe* Mti2 (**c**) and Mti2Δinsertion (**d**). Protein structure predictions were conducted using AlphaFold 3 with the default parameters (**e**). Structure Alignment of Mti2 and Mti2Δinsertion was performed by TM-align [25], which uses a TM-score to quantify the structural similarity. A TM-score ranging from 0 to 0.3 indicates random structural similarity, while scores between 0.5 and 1.0 suggest that the two structures almost share the same fold. In contrast, a higher root mean square deviation (RMSD) value reflects a significant structural difference between two structures. All structural figures were generated using PyMOL (Version 3.0.3, Schrödinger, LLC, New York, NY, USA).

**Figure 2 biomolecules-15-00695-f002:**
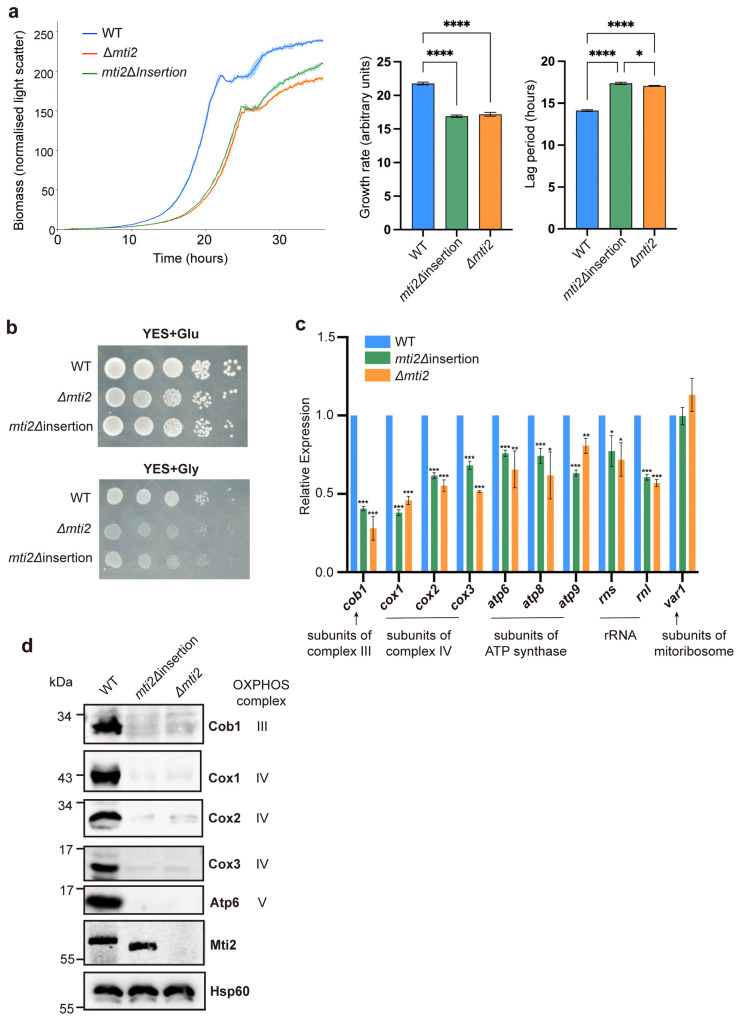
Deletion of the insertion domain impairs Mti2 function. (**a**) Left graph: Wild-type, *mti2*Δ*insertion* and Δ*mti2* strains were grown in a microbioreactor as described [26], and the mean growth curves are shown, along with the SD (shaded regions) for three independent replicates (using R package *grofit* [27]. Middle and right graphs: Quantitation of the growth rates and lag periods for the assays shown in the left graph. The statistical analysis was conducted by one-way ANOVA, followed by Tukey’s honest significance test (*, *p* < 0.05; and ****, *p* < 0.0001). (**b**) The wild-type (WT), *mti2*Δ*insertion* and Δ*mti2* strains were precultured overnight in YES medium at 32 °C, and the cultures were subsequently diluted to an initial OD_600_ of 0.2. A total of three OD_600_ of the cells were collected after 12 h, and 10-fold serial dilutions were prepared and spotted onto YES solid plates containing 3% glucose (top; fermentative condition) or 3% glycerol (bottom; non-fermentative condition). The plates were incubated at 32 °C for 2 days and photographed. (**c**) qRT-PCR analysis of mtDNA-encoded genes in wild-type, *mti2*Δ*insertion* and Δ*mti2* cells; the levels of mt-RNAs in *mti2*Δ*insertion* and Δ*mti2* cells were normalized to *act1* mRNA and measured as fold changes relative to the wild-type cells set to 1. All reactions were performed in triplicate, and the statistical significance was determined by the Student’s *t*-test using GraphPad Prism software (*, *p* < 0.05; **, *p* < 0.01 and ***, *p* < 0.001). (**d**) Western analysis of mtDNA-encoded proteins from mitochondrial extracts using corresponding antibodies. The nuclear-encoded Hsp60 served as the loading control.

**Figure 3 biomolecules-15-00695-f003:**
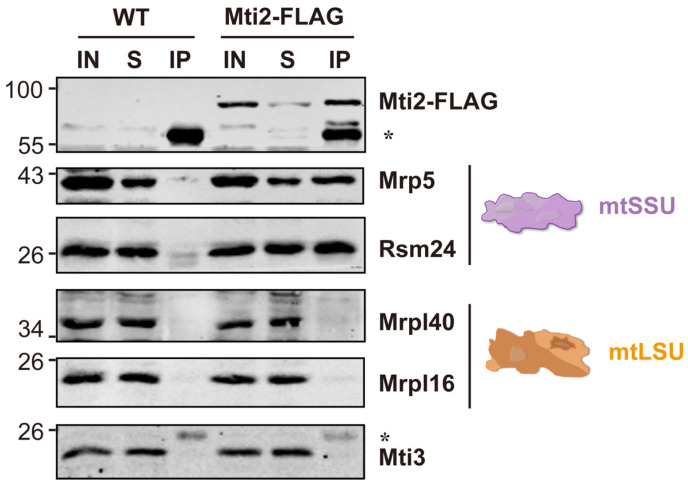
Mti2 physically interacts with small subunits of mitochondrial ribosomes. Homogenized cultures of wild-type and Mti2-FLAG strains were subjected to anti-FLAG co-immunoprecipitation. Mitochondrial extracts (IN), co-IP supernatant (S) from incubation after centrifugation and immunoprecipitates (IPs) were analyzed by Western blotting using specific antibodies against mitochondrial translation initiation factors (Mti2 and Mti3), small subunits (Mrp5 and Rsm24) and large subunits (Mrpl40 and Mrpl16) of mitochondrial ribosomes. The asterisk indicates an unspecific band. The CoIP assay was independently repeated three times.

**Figure 4 biomolecules-15-00695-f004:**
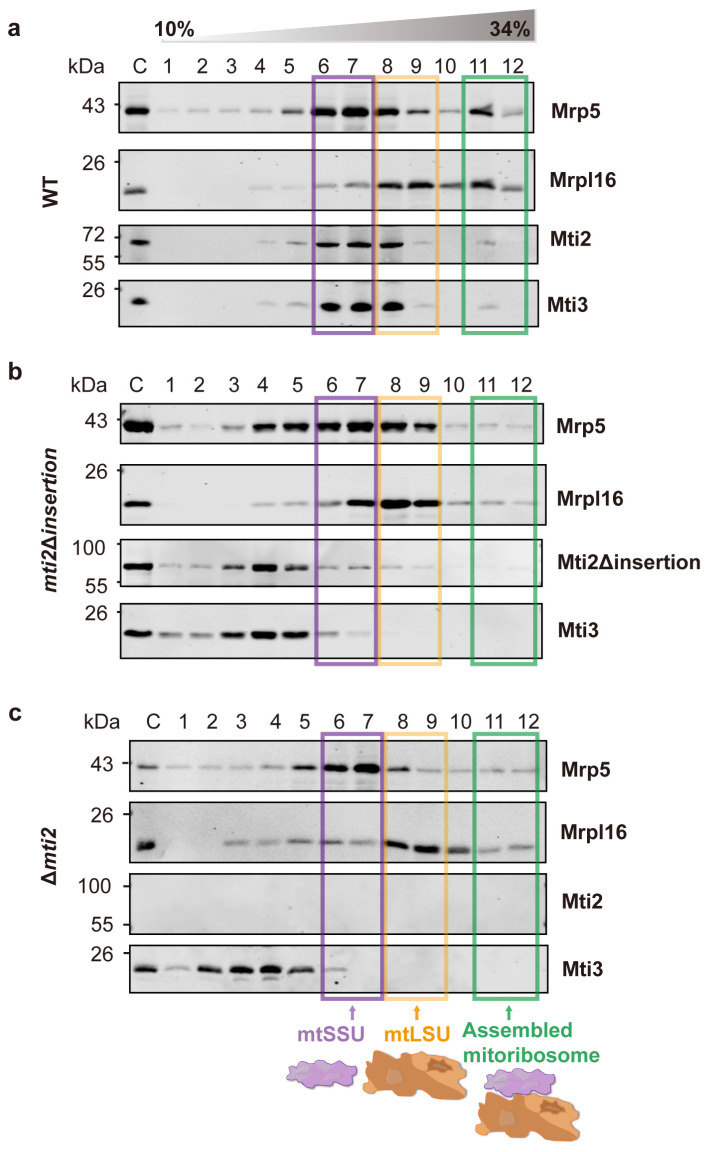
Deletion of the insertion domain reduces the affinity between the translation initiation factors and mtSSU. Sucrose sedimentation analysis of the mitochondrial translation initiation factors across mitoribosomal subunits and fully assembled mitoribosomes in the wild-type (**a**), *mti2*Δ*insertion* (**b**) and Δ*mti2* (**c**) strains. Crude mitochondria from the different strains were isolated, lysed and loaded onto a 10–34% sucrose gradient for ultracentrifugation. Twelve fractions were collected from the bottom (fraction 12) to the top (fraction 1), with C indicating the input mitochondrial protein control. Proteins were precipitated and analyzed by Western blotting using specific antibodies against the mitochondrial translation initiation factors (Mti2 and Mti3), the small subunit (mtSSU, Mrp5) and the large subunit (mtLSU and Mrpl16) of mitoribosomes. Transparent purple, orange and green boxes mark the peak fractions of the mtSSU, mtLSU and assembled mitoribosomes, respectively. The experiment was independently repeated three times.

## Data Availability

The original contributions presented in this study are included in the article/Appendix A. Further inquiries can be directed to the corresponding authors.

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
