# Peer review of "The Insertion Domain of Mti2 Facilitates the Association of Mitochondrial Initiation Factors with Mitoribosomes in Schizosaccharomyces pombe"

_biomolecules, 2025, doi:10.3390/biom15050695_

Round 1

Reviewer 1 Report

Comments and Suggestions for Authors

Mitochondria contain their own DNA (mtDNA), which codes for a very limited number of proteins involved in the expression of mitochondrial OXPHOS genes. Most of the proteins involved in mitochondrial protein synthesis are encoded by nuclear DNA, synthesised in the cytoplasm and imported into the mitochondria. Initiation of translation in mitochondria also involves unique mechanisms, distinct from those in the cytosol or bacteria. For example, for mitochondrial translation initiation, the initiation factor IF1 is absent and the IF2 factor (Mti2) has an additional domain already studied in the equivalent factor in human mitochondria (MTIF2). The present work focuses on the Schizosaccharomyces pombe Mti2 factor, which is the ortholog of human MTIF2.

Using bioinformatics analysis and structure prediction (Alphafold3), the authors provide data suggesting that the insertion domain is essential for correct folding. Deletion of the insertion domain reduces the growth rate of cells with a prolonged lag phase. A clear growth deficit on glycerol medium reveals a defect in mitochondrial respiration. The deletion also led to a decrease in the mRNA levels of the mitochondrial subunits of the respiratory complex and almost complete disappearance of the corresponding proteins in the mitochondria. Using co-IP, the authors showed that Mti2 interacts with the small subunit of the mitoribosome (mtSSU) but not with Mti3, the second mitochondrial translation initiation factor. Finally, the authors analysed the interaction of Mti2 and derivatives deleted from the insertion domain with the mitoribosomes and subunits by sucrose gradient sedimentation. The results showed that deletion of the insertion domain leads to dissociation of Mti1 and Mti2 from the small subunit (mtSSU). In summary, the authors demonstrate the conserved and essential role of the S. pombe Mti2 insertion domain in mitochondrial protein synthesis, as does the human orthologue MTIF2.

Overall, the article is well written and presented, but some experimental details are missing. The results focus mainly on interactions and the function of Mti2 is not directly studied, and the proposed conclusions are often based on the results obtained with MTIF2.

A specific point of concern relates to the analysis by sedimentation on a sucrose gradient. The authors should provide details of the experimental protocol. What type of rotor did they use? An absorbance profile at 260 nm should be presented to compare sedimentation in the different samples. For example the sample in 4B (mit2∆insertion) shows a different separation based on the distribution of Mrp5 and Mrpl16 which is quite different from the WBs in 4A and 4C. It looks like the sample has sedimented less and therefore the coloured boxes should be shifted to the left. This would lead to a different conclusion, i.e. that Mti2∆ins and Mti3 interact with mtSSU.

How many times was this sucrose gradient sedimentation experiment repeated? The same goes for the Co-IP experiments and other experiments.

Reviewer 2 Report

Comments and Suggestions for Authors

Authors present the study performed using Schizoaccharomyces pombe cells showing importance of insertion domain of mitochondrial translation factor Mti2 in mitochondrial synthesis of five mitochondrially encoded proteins and for cell growth on medium with glycerol, non-fermentable carbon source. This domain is also needed for interaction of Mti2, together with other factor Mti3, with small subunit of mitoribosome. This documents that in yeast, similarly as in humans, Mti2 with variable insertion domain compensates for the absence in mitochondria of initiation factor 1, IF1, which is present in cytoplasmic and bacterial translation systems. Findings are novel, but not very innovative, similar results were already shown in other organism. Results are well described, however, require corrections.

General comments

In Discussion the results of Fig.2c are not discussed. Is not clear why this experiment was performed. How these results correspond with respective protein levels?

Specific comments

P5, L193 “ the P site” of what? What is P site, what about other sites in the ribosome? There is no information in the Introduction about sites in the ribosome and suddenly here is P site. Remove or describe better.

P9, L247 how cells were pregrown, how serial dilutions were made, 10x or 5x?

P9, L252 were normalized to ACT1 mRNA, not actin, the protein

P9,L263 the RNA level of var1, encoding mitochondrial ribosomal subunit, remained stable. var1 is a gene, not the protein.

P9, L 265, 266 Sentence is not correct and not agree with Fig.2d. Fig.2d shows that in mti2Δinsertion mutant the Mti2 protein is present, but migrates faster, and this is in agreement with partial deletion introduced in the MTI2 gene. Only in mti2Δ cell the Mti2 protein is not observed, as expected for full deletion of the gene.

P10, L 300 rather prevents association of translation factors Mti2 and Mti3 with mtSSU, not resulted in dissociation.

P12, L325 explain what is Shine-Dalgarno sequence or remove.
